# Motion Artifacts Reduction for Noninvasive Hemodynamic Monitoring of Conscious Patients Using Electrical Impedance Tomography: A Preliminary Study

**DOI:** 10.3390/s23115308

**Published:** 2023-06-03

**Authors:** Thi Hang Dang, Geuk Young Jang, Kyounghun Lee, Tong In Oh

**Affiliations:** 1Department of Medical Engineering, Graduate School, Kyung Hee University, Seoul 02453, Republic of Korea; 2018310929@khu.ac.kr; 2Department of Thoracic and Cardiovascular Surgery, Seoul National University Bundang Hospital, Seongnam-si 13620, Republic of Korea; rmrdud@gmail.com; 3Medical Science Research Institute, Kyung Hee University Medical Center, Seoul 02447, Republic of Korea; imlkh84@gmail.com; 4Department of Biomedical Engineering, School of Medicine, Kyung Hee University, Seoul 02453, Republic of Korea

**Keywords:** electrical impedance tomography, motion artifact detection, heart rate, cardiac output, hemodialysis, source consistency

## Abstract

Electrical impedance tomography (EIT) can monitor the real-time hemodynamic state of a conscious and spontaneously breathing patient noninvasively. However, cardiac volume signal (CVS) extracted from EIT images has a small amplitude and is sensitive to motion artifacts (MAs). This study aimed to develop a new algorithm to reduce MAs from the CVS for more accurate heart rate (HR) and cardiac output (CO) monitoring in patients undergoing hemodialysis based on the source consistency between the electrocardiogram (ECG) and the CVS of heartbeats. Two signals were measured at different locations on the body through independent instruments and electrodes, but the frequency and phase were matched when no MAs occurred. A total of 36 measurements with 113 one-hour sub-datasets were collected from 14 patients. As the number of motions per hour (MI) increased over 30, the proposed algorithm had a correlation of 0.83 and a precision of 1.65 beats per minute (BPM) compared to the conventional statical algorithm of a correlation of 0.56 and a precision of 4.04 BPM. For CO monitoring, the precision and upper limit of the mean ∆CO were 3.41 and 2.82 L per minute (LPM), respectively, compared to 4.05 and 3.82 LPM for the statistical algorithm. The developed algorithm could reduce MAs and improve HR/CO monitoring accuracy and reliability by at least two times, particularly in high-motion environments.

## 1. Introduction

Electrical impedance tomography (EIT) was developed for simultaneous, noninvasive monitoring of cardio-pulmonary functions regardless of intubation or the patient’s conscious status [1,2,3]. Since air volume in the lungs and blood volume in the heart alter the electrical impedance measured in the thorax over time, thoracic time-difference EIT images present the functional changes in the lungs and heart between two different time points [4]. The images are reconstructed from boundary voltages measured by multiple electrodes attached to the thorax surface. Therefore, the quality of EIT images highly depends on the stable and well-contacted electrode–skin interface [5]. However, changes in electrode-skin impedance caused by posture changes, talking, sighing, coughing, and other motions induce numerous artifacts in measured data [6,7]. These artifacts distort information in the reconstructed images or extracted diagnostic parameters.

Nevertheless, the respiratory-related volume signal (RVS) and cardiac-related volume signal (CVS) separated from the time-series of EIT images on the thorax have the advantage of real-time monitoring of hypoventilation in non-intubated patients with respiratory depression and managing patients with an unstable hemodynamic state, simultaneously [8,9,10]. Specifically, noninvasive hemodynamic monitoring is important for patients undergoing hemodialysis (HD) because the hemodynamic imbalance induced by HD leads to significant stress on the heart and peripheral vasculature [11,12,13,14]. In addition, real-time hemodynamic monitoring could improve the patient’s safety and clinical efficiency of HD by guiding appropriate interventions for cases of hypertension or hypotension because many patients undergoing HD have cardiovascular diseases [11,13]. However, one of the challenges of using EIT to monitor the hemodynamics of conscious patients during HD is motion artifacts (MAs) because CVS is relatively smaller than RVS in amplitude and very sensitive to motion [15]. In particular, the patient’s hemodynamic state changes rapidly during the HD treatment of about 4 h, and the introduction of nonperiodic movements significantly degrades the measurement quality. Although various signal processing methods could be applied to suppress such artifacts, they often do not discard artifact-contaminated data, leading to the generation of false-positive warning cases in respiratory or hemodynamic monitoring [8,9,10,16].

CVS and photoplethysmography (PPG), although measured using different sensors at different positions, have similar patterns because both reflect changes in blood volume. Therefore, several conventional methods used to detect and reduce MAs in PPG can be candidates to adopt for CVS [17,18,19,20]. For example, the signal sections contaminated by MAs could be detected by tracking rapid changes in the amplitude of the accelerometer attached to the subject [19,21,22]. However, in many situations without available motion sensors, MAs could still be detected using variable frequency complex demodulation (VFCDM), wavelet transforms (WT), empirical mode decomposition (EMD), and a statistical approach. The VFCDM could detect MAs by detecting the different spectral characteristics of the noise from the clean signal [23]. The WT could be applied to decompose the signal into various frequency bands and detect components with high-frequency noise [24,25]. The concept of EMD was decomposing the signal into a set of intrinsic mode functions (IMFs) by analyzing each IMF’s amplitude and frequency characteristics. It could detect components affected by MAs [26]. Nevertheless, the common limitation of VFCDM, WT, and EMD was the computational complexity, which limited these methods from practical implementation for real-time applications. The statistical method observes changes in the amplitude and morphology of cardiac cycles through the statistical parameters of standard deviation, kurtosis, skewness, and so on [27]. Without the appearance of MAs, the amplitude and morphology of the CVS in the cardiac cycle are repeated with the same pattern. Therefore, the statistical parameter converges to a constant over time. By setting thresholds based on statistical values obtained from data without MAs and comparing them with those calculated in upcoming cardiac cycles, data sections affected by MAs could be detected and either discarded or corrected [17,27]. However, the performance of this method may be limited by a large amount of noise, since the original signal varies over time. This problem is particularly relevant in patients undergoing HD, who frequently produce severe motions because of pain and long treatment, highlighting the need for alternative approaches for MA detection and correction in these specific circumstances. The previous works have evaluated the performance of developing algorithms on simulation data or measurements from healthy subjects, which are usually very different from ill patients. In addition, most of the attempted MA reduction algorithms are based on frequency analysis. Thereby, these methods may be less effective for detecting types of MAs with distortions in the signal shape.

This study proposes a new algorithm to detect and correct MAs in the CVS for monitoring the HR and CO of conscious patients. Electrocardiogram (ECG) monitoring is usually performed in clinical measurements of suspected cardiovascular disease patients. Since ECG and CVS are generated by one heart beating, they should have the same frequency and phase matching. Furthermore, they would have different artifacts since CVS and ECG signal are obtained from two independent devices and electrodes at different locations on the body. Therefore, source-consistent CVS and ECG signal characteristics could help to detect and correct MAs for monitoring hemodynamic state. The proposed algorithm simultaneously considered the features of signal morphology and frequency to reduce MAs in clinical applications of consciously ill patients. This approach differs from the methodologies of previous studies that reduced MAs solely by relying on one feature, often utilizing simulation data or measurements from healthy subjects.

The objectives of this paper were to (1) build a strategy to detect and correct MAs in CVS with source consistency of ECG and CVS and (2) evaluate its usefulness in improving heart rate (HR) and cardiac output (CO) monitoring from clinical data. First, we explain how to collect clinical data from subjects during HD and extract hemodynamic parameters and motion data from the measured data. After a detailed description of the proposed algorithm, performance parameters for comparing algorithms in the HR and CO monitoring and statistical analysis methods are presented. Finally, experimental data are analyzed and compared using the performance parameters presented according to the range of motion index.

## 2. Materials and Methods

### 2.1. Materials for Data Collection

Thirty-six measurements were conducted on fourteen patients who underwent HD treatment at Kyung Hee University Hospital in Gangdong, Republic of Korea. We recruited subjects (18 to 80 years) who had experienced intradialytic hypotension for the last 3 months. Severely obese patients with a body mass index over 50, pregnant or baby-feeding patients, patients with a history of respiratory diseases including chronic obstructive pulmonary disease and moderate to severe asthma, patients with implantable electronic medical devices, or patients with insufficient contact of electrodes to the skin were excluded. The institutional review board at Kyung Hee University Hospital approved the clinical study protocols on 3 July 2020 (KHNMC2020-08-006). Written informed consent has been obtained from all patients. During HD, EIT data was recorded by AirTom^TM^ (BiLab, Seongnam-si, Republic of Korea) with a temporal resolution of 100 frames/s along with a continuous single-lead ECG, as shown in Figure 1. CO was measured every 20 s using an EV1000^TM^ with a ClearSight finger cuff (Edwards Lifesciences, Irvine, CA, USA) for obtaining reference data. This device is currently used in clinics as a solution for noninvasive hemodynamic monitoring, even though it is susceptible to the position of the sensor and motion noise [28]. Additionally, an inertial measurement unit (IMU) was attached to the disposable electrode pad (Epad^TM^, BiLab, Seongnam-si, Republic of Korea) to monitor the posture and motion of patients. The IMU (WT901C, Witmotion, Shezhen, China) produces data on 3-axis accelerations, 3-axis angular velocity, and 3-axis angles. During measurements, every motion caused by the patient was observed, such as body movement, talking, hand moving, coughing, and so on, and recorded in a case log sheet.

### 2.2. Hemodynamic Parameters and Motion Index

The CVS was extracted from EIT images by successive applications of principal component analysis (PCA) and independent component analysis (ICA) [10,29]. An additional rule was applied for selecting the ICA component with the most similar frequency and phase synchronization to the measured ECG signal. The CVS was automatically extracted proportionally to the change in blood volume due to the heartbeat. The customized peak detection algorithm was applied to detect valleys and peaks in CVS waveforms [10]. Cardiac cycles were determined as the section between two valleys in the CVS. The estimated stroke volume (SV) was directly proportional to the valley-to-peak values in the CVS. The SV measured by EIT (SV_EIT_) was calibrated by multiplying it with a scale factor calculated by the ratio of SV measured by EV1000^TM^ (SV_EV1000_) and measurements from EIT taken simultaneously before starting HD. The Pan–Tompkins algorithm was used to find R-peaks in the ECG signal [30]. Then, the number of beats and cycles were counted during each minute to calculate heart rate (HR_ECG_ and HR_EIT_) from ECG and CVS, respectively. The CO was calculated as follows: CO = HR × SV. Each interbeat interval (IBI_EIT_, IBI_ECG_) was defined by the distance between two adjacent peaks in the CVS and ECG signal, as shown in Figure 2.

During the HD start and end time, medical staff were performing multiple treatments and moving the patients, so data for 5 min after starting and 5 min before finalizing the HD were excluded from the analysis. Each measurement was divided into sub-datasets for one hour. Each sub-dataset contained 4500 cardiac cycles on average. A total of 113 one-hour datasets were randomly split into training and testing subsets. The split ratio was 2:8. The 22 training subsets were used to design the new algorithm and the 91 testing subsets were used for evaluation.

The nine signals measured by the IMU were each passed through a low-pass filter with a 3 Hz cutoff frequency to obtain motion and posture information. Then, the standard deviation of the data was calculated while moving a 1 s window [16]. Finally, MAs were detected based on the thresholds calculated in the movement-restricted section before starting HD. Compared with motions written in the case log sheet during HD, most large MAs were detected using IMU data. A motion index (MI) was defined as the number of detected motions per one-hour sub-dataset. All data sections were classified into five groups based on the MI range: 0–5, 5–10, 10–20, 20–30, and >30.

### 2.3. Real-Time Motion Artifacts Reduction Algorithm Based on Source Consitency

This study proposes an algorithm to detect MAs in the CVS, based on the source consistency of ECG and CVS, which are directly related to changes in heartbeat and blood volume. Since each CVS and ECG signal was recorded through independent devices and electrodes, both sets of data may have different measurement and artifact features but should have the same frequency and phase matching. Therefore, MAs in the CVS can be detected from discrepancies between the ECG and CVS. When an MA occurs, the quality of both the CVS and ECG signal is degraded, or the amplitude, frequency, and phase of one signal suddenly change so that the phase and frequency of the two signals are no longer matched.

Figure 2 illustrates an example of 12 s of data, including CVS extracted from EIT images, the ECG signal, and the motion signals of acceleration, angular velocity, and angle measured from the IMU. The blue region represents the contaminated MA section of the CVS, as defined by the method using the IMU sensor’s outputs. The phase and time duration between the two independently measured signals of ECG and CVS were unmatched, and the SVs also strongly fluctuated. Additionally, more than one CVS peak existed within one extracted ECG cycle. However, during most periods outside the MA-detected span, the phase and time duration between IBI_EIT_ calculated by CVS and IBI_ECG_ calculated by ECG coincided, and the SVs were stable. Based on these observations, a new MA detection and reduction strategy was established, as displayed in Figure 3. Data during one cardiac cycle is determined as good quality if it fulfills the following four conditions:#Peak_EIT_ = 1 within 1 cardiac cycle in ECG,(1)
(2)$$|{\mathrm{IBI}}_{\mathrm{EIT}}-{\mathrm{IBI}}_{\mathrm{ECG}}|\;\text{<}\;\alpha ,$$
(3)$$|∆{\mathrm{SV}}_{\mathrm{EIT}}|\;\text{<}\;\beta \times {\text{SV}}_{\text{ref}},$$
(4)$${\mathrm{SV}}_{\mathrm{EIT}}\;\in \;[\gamma_{1}\times \;{\mathrm{SV}}_{\mathrm{ref}},\gamma_{2}\times {\mathrm{SV}}_{\mathrm{ref}}],$$
where #Peak_EIT_ is the number of CVS peaks in the period of two adjacent R-R peaks of ECG, and |IBI_EIT_ − IBI_ECG_| is the time difference between IBI_EIT_ and IBI_ECG_. Criteria (1) and (2) are based on phase and frequency matching between CVS and ECG signal, respectively. The |∆SV_EIT_| is the difference between two adjacent SV_EIT_, calculated by CVS. Criteria (3) and (4) are based on the stability and continuity of the blood volume change, respectively. In the above conditions, the α,β, and $\gamma_{1-2}$ are threshold values obtained from training datasets. However, since rapid SV changes are possible during HD, the threshold values were updated using SV_ref_. SV_ref_ is the median value of SV_EIT_ measured during the most recent 3 min. Each cardiac cycle is classified as MA when at least one of the above four conditions is not satisfied. In Figure 2, the section marked with thick dotted lines, including the blue region, is the MA section detected by the proposed algorithm. The processed data were used to calculate HR_EIT_ and CO_EIT_.

In order to find the optimal thresholds of α,β, and $\gamma_{1-2}$, 96,871 cardiac cycles were used in the training subsets. Each threshold value was adjusted to maximize sensitivity and specificity when applying a new algorithm, compared to the results from IMU sensors. The set of thresholds was considered optimal when the sum of sensitivity and specificity was highest and the specificity was higher than 80% [31]. Here, the sensitivity and specificity were calculated by the ratio of TP/P and TN/N, respectively. IMU sensors determined the P or N of each cardiac cycle based on the presence of motion noise. TP and TN were the data correctly classified by the new algorithm among P and N groups.

### 2.4. Statistical Analysis of Motion Artifact Reduction in Heart Rate and Cardiac Output Monitoring

The aim of this study was the more robust and stable noninvasive monitoring of HR and CO in HD patients whose hemodynamic status changes during treatment by reducing MAs in EIT data. Therefore, the performance of the proposed algorithm and the conventional statistical method were evaluated in the HR and CO measurement process. The study used 423,305 cardiac cycles in the testing subsets. After applying each algorithm, the similarity between HR_ECG_ from ECG and HR_EIT_ from EIT was compared through Pearson’s correlation and Bland–Altman analysis. The 95% limits of agreement (LOA) include both systematic (bias) and random error (precision) for comparing differences between results measured by the two methods. Here, the bias and maximum precision were indicated by the mean and 1.96 × standard deviation (SD) of |HR_EIT_ − HR_ECG_|, respectively.

CO was measured noninvasively using EV1000^TM^ in the clinic. After applying each algorithm to the EIT, the similarity between CO_EIT_ and CO_EV1000_ was compared using Bland–Altman analysis. Similar to the HR comparison, the same equation was used to calculate the maximum precision of LOA between CO_EIT_ and CO_EV1000_, where SD was the standard deviation of |CO_EIT_ − CO_EV1000_|. Additionally, polar plot analysis was used to evaluate the trending ability of CO monitors and the magnitude bias between CO_EIT_ and CO_EV1000_. This study utilized the polar plot analysis rather than the 4-quadrant plot analysis for two main reasons. First, the polar plot can more effectively delineate the spread of data points and identify potential outliers, which can help identify data problems, such as MAs or the performance of the monitoring devices. Second, the 4-quadrant plot mainly focuses on the direction of the change and does not provide information on the magnitude of the difference between the two methods. Instead of displaying the 4-quadrant plot, the concordance rate (CR) was investigated to assess the overall agreement between CO_EIT_ and CO_EV1000_. The CR was the percentage of data points that fall within the upper right and lower left quadrants, divided by the total number of data points [32].

The radius of the polar plot represents the value of m∆CO, calculated as m∆CO = (∆CO_EV1000_ + ∆CO_EIT_)/2, where ∆CO_EV1000_ and ∆CO_EIT_ were the differences between two successive CO_EV1000_ and CO_EIT_, respectively. The exclusion zone was set where ∆CO was less than 0.5 LPM, in accordance with a previous study [33]. The radial limits of agreement (radial LOA) were calculated by the maximum polar angle, containing 95% of the data. Conversely, the inclusion rate (IR) was defined by the proportion of data within the radial sector of ±30°, since other invasive CO monitoring methods resulted in a radial LOA of 30° in the previous study. If the IR is higher than 95%, the CO monitoring method is acceptable to use for clinical purposes in agreement with the trending ability. The polar plot analysis reveals the radial LOA, IR, and m∆CO between CO_EV1000_ and CO_EIT_.

For each one-hour sub-dataset, the correlation coefficient and maximum precision in LOA between HR_EIT_ and HR_ECG_ and the radial LOA/IR/m∆CO/CR between CO_EV1000_ and CO_EIT_ were calculated in three conditions: (1) before MA reduction, (2) after MA reduction using the new algorithm, and (3) after MA reduction using the conventional algorithm. Then, the above quantitative indicators were compared among five MI groups to evaluate the effectiveness of the MAs reduction algorithm, according to the degree of motion, for monitoring HR and CO. The comparisons between indexes were made using the paired *t*-test. The statistical analysis was performed using the MATLAB software (MathWorks, Natick, Massachusetts, USA).

## 3. Results

Datasets, consisting of measurements from the IMU, ECG, and EIT systems collected during HD, were thoroughly evaluated to compare the performance of the new algorithm and the statistical algorithm for MA detection and correction. From the 36 datasets, a total of 113 one-hour sub-datasets were used in the study. When the performance of the proposed MA detection algorithm was tested using 91 sub-datasets, it had a sensitivity of 66.6 ± 2.83% and a specificity of 85.7 ± 1.54% regarding the MAs detected by the IMU sensor. In contrast, the statistical algorithm for MA detection had a sensitivity of 55.5 ± 2.73% and a specificity of 85 ± 1.81% regarding the MAs detected by the IMU sensor. The new algorithm demonstrated a statistically significant improvement in the sensitivity of MA detection compared to the statistical algorithm (*p* = 0.006). However, there was no statistically significant difference in specificity between the two methods (*p* = 0.78).

Figure 4 shows an example of the results before and after applying the proposed MA reduction algorithm for monitoring HR and CO. It describes a measurement case (measurement #3 from patient #2) consisting of four one-hour sub-datasets. Figure 4a presents the scatter and Bland–Altman plots between HR_EIT_ and HR_ECG_. The Pearson correlation coefficient was increased from 0.96 to 1.0, and the maximum precision of LOA was reduced from 1.7 to 0.46 beats per minute (BPM) after reducing the MAs. In addition, the root mean square error (RMSE) between HR_EIT_ and HR_ECG_ was decreased from 0.87 to 0.22 BPM, where RMSE was the ratio between the sum of squared errors and the number of heartbeats. Figure 4b shows Bland–Altman and polar plots between CO_EIT_ and CO_EV1000_ from the same measurement. The solid and dot lines in the Bland–Altman plot represent the bias and 95% LOA, respectively. The solid and dot lines in the polar plot show the ±30° angles and radial LOA, respectively. After reducing MAs, the maximum precision of LOA decreased from 2.0 to 1.4 L per minute (LPM), and the upper limit of m∆CO reduced from 7.05 to 1.97 LPM. The IR and CR were increased from 93% to 97% and 90% to 94%, respectively, when applying the proposed MA reduction algorithm.

In the testing data, 91 one-hour sub-datasets were divided into 5 groups based on the range of MI: 0–5, 5–10, 10–20, 20–30, and >30. The number of one-hour sub-datasets classified into the 5 groups were 13, 19, 21, 24, and 14, respectively. Before applying the MA reduction algorithm, the Pearson correlation coefficient between HR_ECG_ and HR_EIT_ was 0.68, and the maximum precision of LOA was 3 BPM when the MI was lower than 30 (MI < 30). In contrast, when the MI exceeded 30, the correlation coefficient rapidly decreased to 0.4, and the maximum precision of LOA increased to 5.2 BPM. Figure 5 presents the comparison of the correlation coefficient and the maximum precision of LOA between HR_ECG_ and HR_EIT_ according to the degree of MI after applying the proposed and statistical MAs reduction algorithms, respectively. The new algorithm produced a higher performance of correlation and precision than the conventional statistical algorithm in all MI ranges. When the MI was less than 30, the new algorithm and statistical algorithm produced a correlation of 0.93 and 0.83 on average, respectively. As the MI increased beyond 30, the correlation coefficient from the new algorithm was reduced to 0.83, whereas the correlation coefficient with the statistical algorithm dropped sharply to 0.56. In the Bland–Altman analysis, similar results were observed when comparing the two algorithms. The maximum precision of LOA between HR_ECG_ and HR_EIT_, after reducing MAs with the new algorithm, was 1.06 BPM at MI < 30 and slightly increased to 1.65 BPM at MI > 30. However, when reducing MAs with the statistical algorithm, the maximum precision of LOA was 1.68 BPM at MI < 30 and significantly increased to 4.04 BPM at MI > 30. The proposed algorithm produced a smaller precision of LOA than the statistical algorithm, particularly at higher MI values.

Before applying the MA reduction algorithm to the same 91 one-hour sub-datasets, the maximum precision of LOA in the Bland–Altman plot, radial LOA, IR, the upper limit of m∆CO from the polar plot, and CR between CO_EV1000_ and CO_EIT_ were 7 LPM, 49.8°, 93.1%, 10 LPM, and 81.4%, respectively, at MI < 30. When the MI was increased to over 30, they changed to 20 LPM, 59°, 86.7%, 30 LPM, and 68.9%, respectively. After applying the proposed and statistical algorithms to reduce MAs, Figure 6 shows the differences in the maximum precision of LOA and the upper limit of m∆CO between CO_EV1000_ and CO_EIT_ according to the degree of MI. The new algorithm produced a better precision of LOA and an upper limit of m∆CO compared to the conventional statistical algorithm. When the MI < 30, the maximum precision of LOA and upper limit of m∆CO with the new algorithm were 2.4 LPM and 2.2 LPM, respectively. These values increased to 3.41 LPM and 2.82 LPM, respectively, as the MI exceeded 30. In contrast, the statistical algorithm produced a maximum precision of LOA and upper limit of m∆CO of 2.5 LPM and 2.8 LPM, respectively, at MI < 30, which increased to 4.05 LPM and 3.82 LPM, respectively, at MI > 30. The difference in performance between the two methods was most pronounced at MI > 30.

The radial LOA, an index used to evaluate the trending ability of the CO monitor, also represents the performance of the algorithm according to the degree of MI. When the MI was less than 10, the radial LOA of the data processed by the proposed algorithm and the conventional statistical algorithm was 26.95° and 28.6°, respectively, showing a difference of 5.9%. However, when the MI was over 20, the proposed algorithm’s radial LOA rose slightly to 27.6°, whereas the radial LOA of the data processed with the conventional statistical method was 30.6°, showing a difference in performance of 10.3%. The IR and CR between CO_EV1000_ and CO_EIT_ was not different after reducing MAs by either the proposed (95.9 ± 0.48%, 84.9 ± 2.3%) or conventional statistical algorithm (95.4 ± 0.43%, 85.4 ± 2.02%). Table A1 presents a detailed description of the evaluation metrics for HR and CO monitoring before and after applying MA reduction algorithms.

## 4. Discussion

When monitoring the hemodynamic state of a conscious patient for a long time, such as during HD treatment, the patient’s movement significantly affects the measured signal. Furthermore, it is difficult to detect the features of motion noise in the signal being measured in real-time since patients’ motions cannot be restricted. In particular, hemodynamic signals dynamically change when removing excess water accumulated in the patient through the blood during HD. Therefore, detecting and correcting MAs in the presence of such changes in the original hemodynamic signal has low sensitivity, and erroneous correction may distort the estimated parameters instead. In this study, based on the source consistency between the electrical signal that generates the heartbeat and the volume change in the blood flow caused by the heartbeat, a new algorithm was proposed for detecting and correcting the MAs of EIT data for patients whose hemodynamic state rapidly changes.

With reference to the results of MA detection using a separate IMU device, the proposed method performed better in HR and CO monitoring than the conventional statistical algorithm. In particular, in periods of excessive motion noise caused by, for example, pain during HD, the new method effectively improved the quantitative metrics of HR and CO monitoring compared to the conventional statistical algorithm. Considering that intradialytic hypotension occurs in approximately 20–30% of patients receiving HD treatment, which is accompanied by unbearable pain to cause a halt of HD, detection and correction of excessive MAs due to unintended pain is highly required [34]. Furthermore, it is important to understand that the patient’s hemodynamic state dramatically changes during HD treatment. Therefore, MA detection and correction methods, other than statistical methods, for morphological changes in signals are considered meaningful.

There was a report that CO can be reduced by up to 30 ± 13% during HD [35]. Since the statistical algorithm only considers changes in amplitude and morphology with constant thresholds, it can lead to possible false positive or false negative warnings, especially when rapid blood volume changes during HD and complex MAs are present. Dynamic threshold values used in the proposed method helped to detect MAs in the specific circumstances of HD treatment. Although the new algorithm was validated through post-processing using measured data, it is fully automated and can operate in real-time without needing an IMU sensor.

The research focused on developing MA reduction techniques using EIT data obtained from consciously ill patients in clinical trials and evaluating the algorithm’s effectiveness in HR and CO monitoring. This kind of study for conscious and non-intubated patients was relatively scarce. In terms of unconscious patients, noninvasive CO monitoring using noninvasive technologies generally achieves a precision of 2.25 LPM. Our results align with this precision when the MI is less than 30 (2.4 LPM), but they deteriorate when the MI exceeds 30 (3.41 LPM) [36]. Braun et al., in a study involving 10 healthy adult volunteers under controlled conditions in a supine position, demonstrated a CR of 80.9% for relative SV changes measured by EIT compared to reference devices. This CR was lower than the average CR observed in our results (83.9%) [16]. However, Braun et al. reported a highly elevated concordance rate of 94.4% for noninvasive SV change measurements in critically ill patients during a fluid challenge (39 measurements, 20 patients), superior to our results [37]. This difference may be attributed to the accuracy of the reference devices utilized and the difference in observing parameters of SV and CO. Furthermore, our experimental results would be degraded by introducing a lot of noise accompanying the pain of HD treatment for conscious, unrestrained patients.

The new algorithm showed a better performance when compared with MAs detected by the conventional statistical algorithm. However, the average sensitivity of the new algorithm for detecting MAs in CVS was 66.6%, which was not a high value. This result was because part of the MA of the CVS was removed in the process of extracting the CVS using ICA. This could be explained by the fact that the number of cardiac cycles with contaminated MAs detected by the CVS was smaller than the number of cardiac cycles detected by the IMU sensor. The CR between CO_EIT_ and CO_EV1000_ was not very good even after applying the proposed algorithm. There was presumably a limitation in the accuracy of the noninvasive CO monitor for the reference measurement, although we improved the accuracy of EV1000^TM^ by using a wrist splint. In summary, the main advantages of the proposed algorithm are that it is easy to apply for real-time MA detection and avoids false positive MA detection generated by an abnormal cardiac activity, because it detects MAs by changes in frequency and morphology simultaneously. Therefore, it can produce better quality HR and CO monitoring in high-motion environments than the conventional algorithm. However, the algorithm has disadvantages, such as the requirement of additional measurements and data synchronization and the performance dependence of the algorithm on the source consistency between the CVS and the other measured signal.

Despite the improved performance of the proposed algorithm, this study had several limitations. First, the quality of the ECG signal plays a crucial role in the performance of the new algorithm. It does not necessarily require a specific ECG signal. Another signal related to the heartbeat and measured by other devices simultaneously with EIT can be used. Fortunately, we can easily obtain high-quality ECG data in the hospital since ECG monitoring is commonly performed for patients either with or suspected of cardiovascular disease. The external signal interface of the EIT system makes it easy to synchronize both data. This function helps to detect phase and frequency matching between the CVS and ECG signals. Although ECG measurement with high accuracy is being performed in clinical practice, additional efforts are needed to secure the quality of the biosignals with source consistency for applying the MAs detection algorithm [38,39]. Furthermore, we would consider adopting artificial intelligence or machine learning techniques to detect MAs in reconstructed EIT images [40,41]. The second limitation was the accuracy of the reference device for CO. The study could not use an invasive CO monitor or echocardiography, which would provide a better performance, because the patients were conscious and repeated measurements were required. Therefore, the EV1000^TM^ was our best choice in the experimental situation. A finger splint was used, and effort was made to keep the position of the hand being measured at the same level as the position of the heart since the quality of the CO_EV1000_ changes were very sensitive to the position and movement of the finger cuff. The third limitation was that the quality of the measured data was highly dependent on the electrode–skin interface unit. The belt-type electrode pad with multiple electrodes attached to the thorax changes the degree of contact by changing the thorax shape and motions. Therefore, a large number of MAs can be generated from small movements due to the low elasticity of the belt and strongly coupled electrode-to-electrode connections. The quality of recorded EIT data could be improved using novel materials for the electrodes, which enhances the contact between the electrode and skin [42,43]. The fourth limitation is that the number of cardiac cycles used to evaluate the proposed algorithm of MA reduction was 423,305, but they were obtained from the 36 measurements of the 14 patients. Still, there is a need to validate the algorithm with data obtained multiple times from various patients. Finally, even though the implemented algorithm can be applied in real-time, the result was processed using it after finishing the data measurement. This raises the need for prospective studies on real-time processing capability validation. Despite these limitations, the ability to monitor HR and CO noninvasively can be of great importance in the real-time display of the status of hemodynamically unstable patients undergoing HD treatment and to intervene and enhance the effectiveness of treatment.

As the use of the EIT system for cardiopulmonary monitoring is expanding toward application to conscious patients who are not connected to a ventilator, such as in post-anesthesia care units and sleep apnea and hypopnea studies, an MA reduction algorithm in CVS is highly demanded. The proposed strategies to detect and correct MAs would help to improve performance in hemodynamic monitoring by EIT. In future work, further prospective clinical studies should be performed with multiple clinical applications to show the effectiveness of the proposed algorithm from data processed in real-time.

## 5. Conclusions

The newly proposed algorithm has shown promising results in reducing MAs and improving HR and CO monitoring accuracy and reliability, particularly in motion-rich environments. This algorithm has been found to be suitable for conscious patients requiring real-time continuous HR and CO monitoring by the noninvasive EIT system. In addition, other measurement combinations may be applied, such as photoplethysmography and EIT, to reduce MAs. However, it is important to note that this algorithm is unsuitable for use in datasets that do not have good-quality signals with source consistency. Additionally, this algorithm can be applied to various EIT medical applications for cardiopulmonary monitoring, such as patient care in the general ward, post-anesthesia care unit, and wearable healthcare, to improve the quality of the measured data.

## Figures and Tables

**Figure 1 sensors-23-05308-f001:**
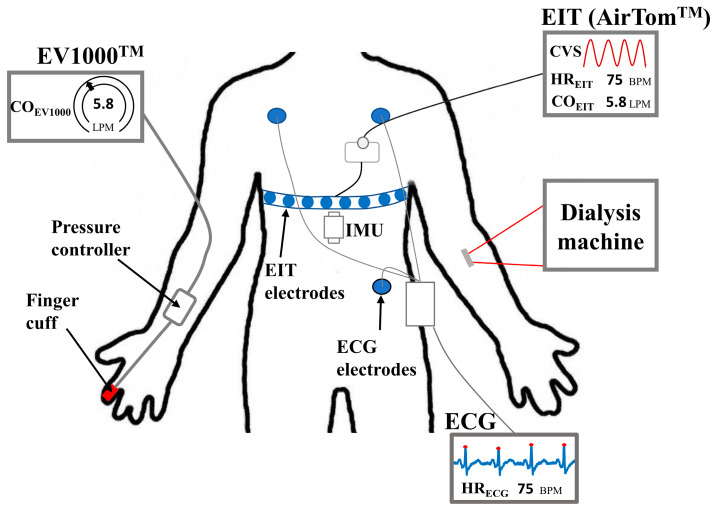
Configuration for data acquisition. EIT data were continuously measured during HD with a single-lead ECG signal through separated electrodes on the thorax. Simultaneously, the patient’s movement and posture were monitored by IMU, and CO was measured at 20-second intervals by an EV1000 ^TM^ with a finger cuff, noninvasively. (EIT: electrical impedance tomography, CVS: cardiac volume signal, IMU: inertial measurement unit, ECG: electrocardiogram, CO_EV1000_: cardiac output from EV1000^TM^, CO_EIT_: cardiac output from EIT, HR_ECG_: heart rate from ECG monitor, HR_EIT_: heart rate from EIT, BPM: beats per minute, and LPM: liters per minute).

**Figure 2 sensors-23-05308-f002:**
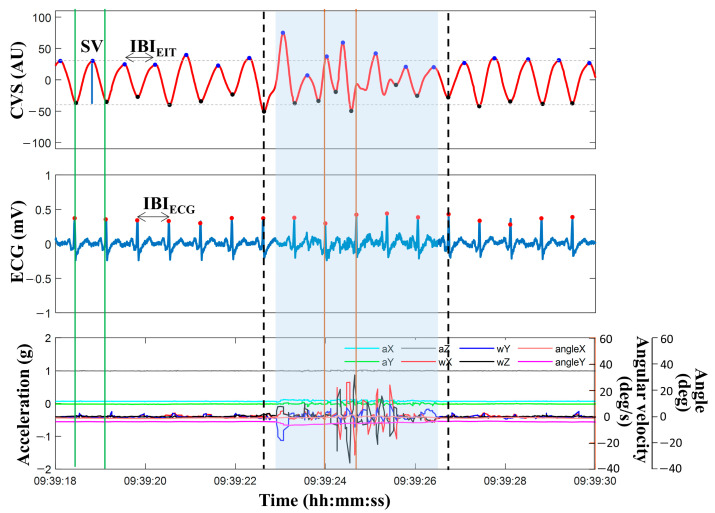
Example of measured data for 12 s, including CVS, ECG, and motion sensing signals from the IMU. The contaminated data region from MAs detected using the IMU sensor was marked with a blue box. The time section marked by dashed lines shows MAs detected by the proposed algorithm. SV was calculated by the amplitude between the peak and valley in the CVS. The IBI_EIT_ and IBI_ECG_ were interbeat intervals in the CVS and ECG, respectively. (CVS: cardiac volume signal, ECG: electrocardiogram, EIT: electrical impedance tomography, SV: stroke volume, IBI_EIT_: interbeat interval from EIT, IBI_ECG_: interbeat interval from ECG, AU: arbitrary unit, deg: degree, and deg/s: degree per second).

**Figure 3 sensors-23-05308-f003:**
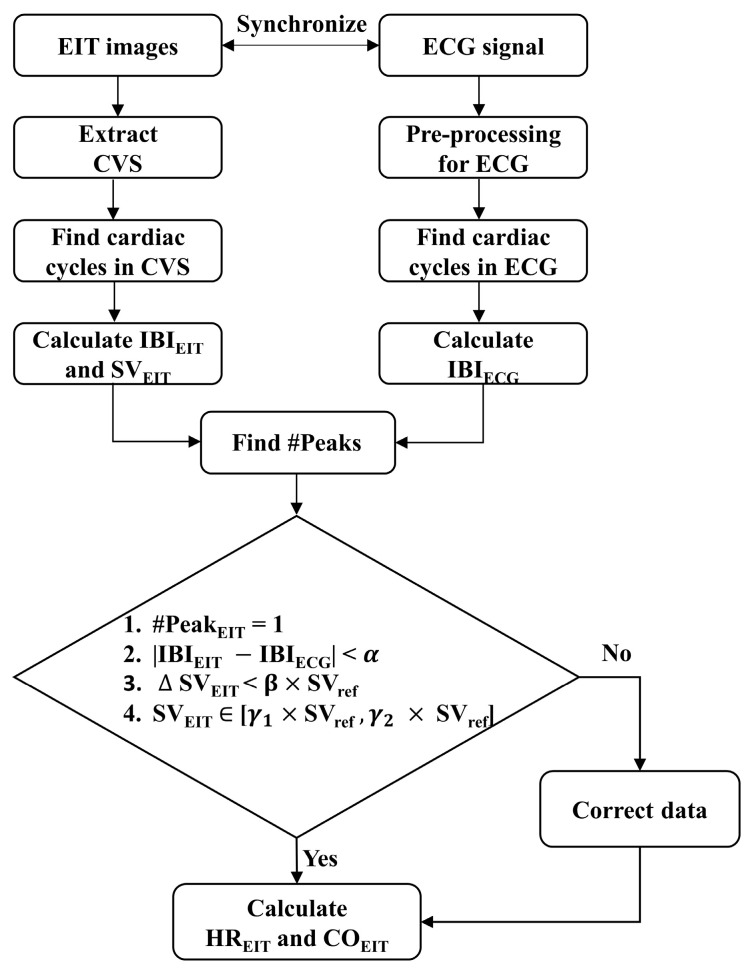
Process of the proposed algorithm for MA reduction based on the source consistency of ECG and CVS. (EIT: electrical impedance tomography, CVS: cardiac volume signal, ECG: electrocardiogram, IBI_EIT_: interbeat interval from EIT, IBI_ECG_: interbeat interval from ECG, SV_EIT_: stroke volume from EIT, SV_ref_: median value of SV_EIT_ during the most recent 3 min, #Peak_EIT_: the number of CVS peaks in the period of two adjacent R-R peaks of ECG, {α,β, and $\gamma_{1-2}$}: set of optimal thresholds from the training subsets, HR_EIT_: heart rate from EIT, and CO_EIT_: cardiac output from EIT).

**Figure 4 sensors-23-05308-f004:**
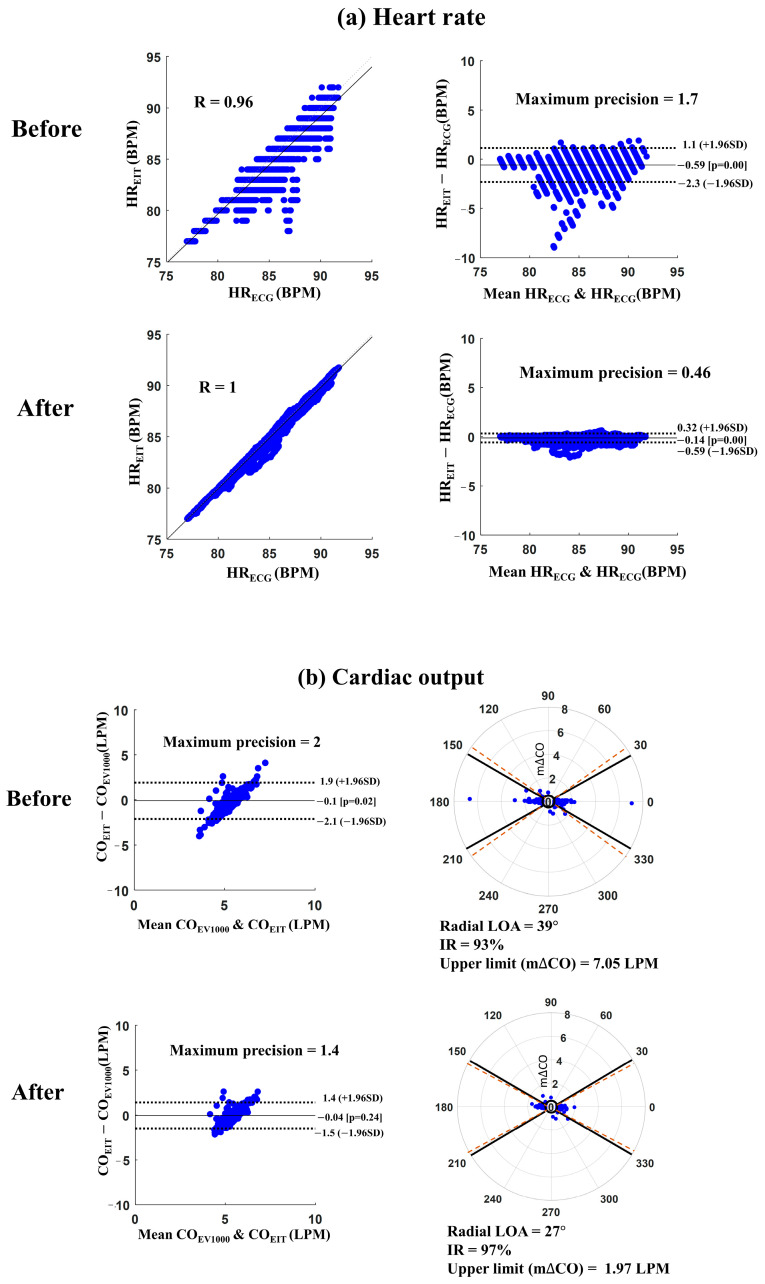
Processed example for measurement #3 from patient #2 before and after applying the proposed MA reduction algorithm for the monitoring of heart rate (HR) and cardiac output (CO). (**a**) Scatter and Bland–Altman plot between HR_EIT_ and HR_ECG_ and (**b**) Bland–Altman plot and polar plot between CO_EIT_ and CO_EV1000_ (HR_ECG_: heart rate from ECG monitor, HR_EIT_: heart rate from EIT, CO_EV1000_: cardiac output from EV1000^TM^, CO_EIT_: cardiac output from EIT, R: Pearson correlation coefficient, Radial LOA: radial limits of agreement, IR: inclusion rate, upper limit (m∆CO): maximum value of mean of CO_EV1000_ and CO_EIT_, SD: standard deviation, BPM: beats per minute, and LPM: liters per minute).

**Figure 5 sensors-23-05308-f005:**
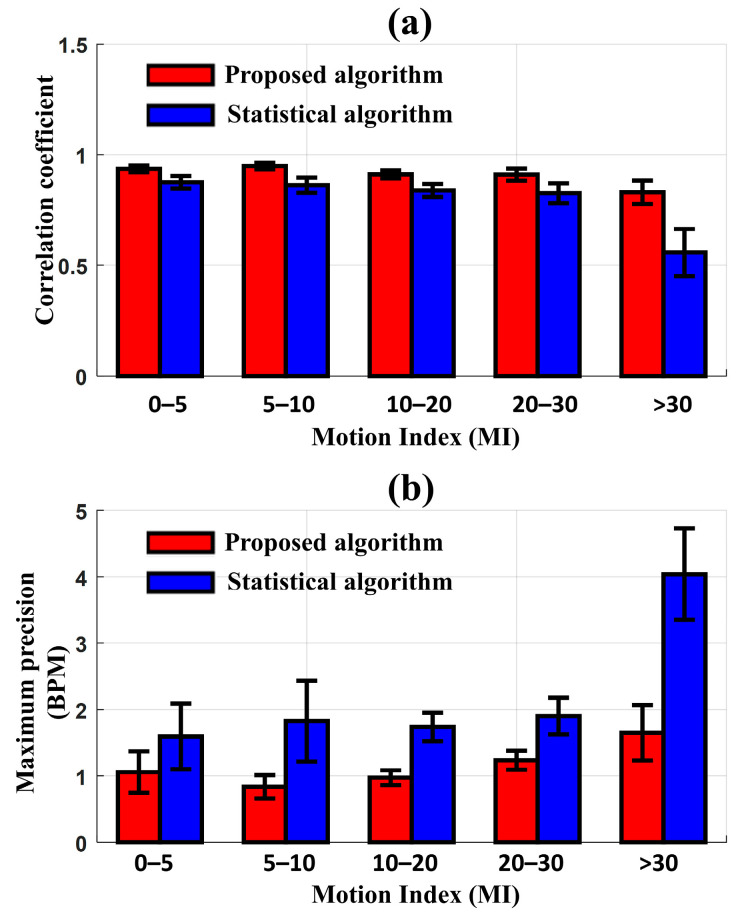
Comparison of the results for the noise reduction algorithms according to the motion index (proposed algorithm [red] and statistical algorithm [blue]). (**a**) Pearson correlation coefficient between HR_ECG_ and HR_EIT_ and (**b**) Maximum precision of the LOA between HR_ECG_ and HR_EIT_ (BPM: beats per minute and MI: number of motions per hour).

**Figure 6 sensors-23-05308-f006:**
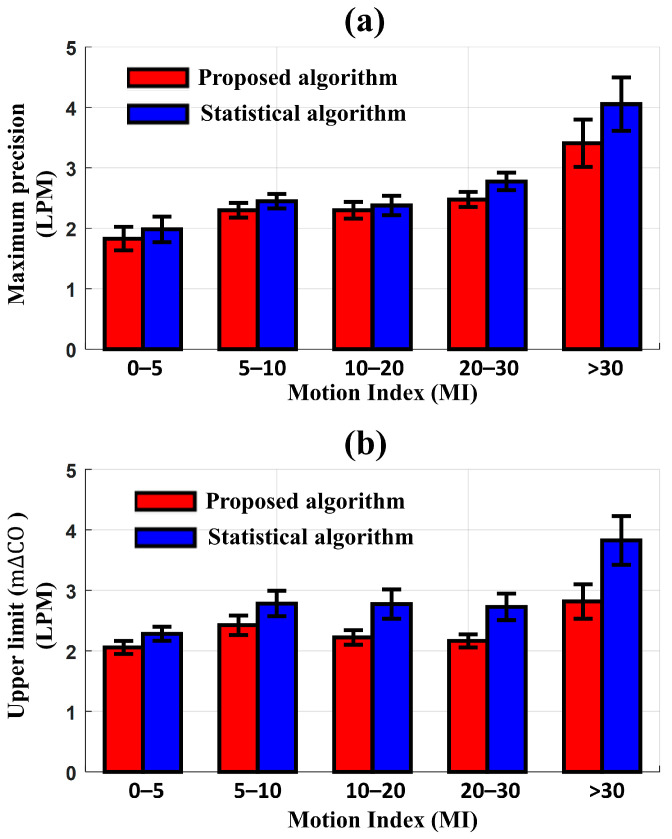
Comparison of the results for the noise reduction algorithms according to the motion index (proposed algorithm [red] and statistical algorithm [blue]). (**a**) Maximum precision of the LOA in Bland–Altman analysis and (**b**) upper limit of m∆CO in polar plot between CO_EV1000_ and CO_EIT_ (LPM: liters per minute and MI: number of motions per hour).

## Data Availability

The data that support the findings of this study are available from the corresponding author, T.I.O., upon reasonable request.

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
