# Peer review of "Motion Artifacts Reduction for Noninvasive Hemodynamic Monitoring of Conscious Patients Using Electrical Impedance Tomography: A Preliminary Study"

_sensors, 2023, doi:10.3390/s23115308_

Round 1

Reviewer 1 Report

Dear Editor and Authors,

I read the paper entitled ‘Motion Artifacts Reduction for Noninvasive Hemodynamic Monitoring of Conscious Patients using Electrical Impedance Tomography’ with great interest.

This study aimed to develop a new algorithm to reduce motion artifacts from the cardiac volume signal for more accurate heart rate and cardiac output monitoring in patients undergoing hemodialysis based on the source consistency between the electrocardiogram and the cardiac volume signal of heartbeats. As the number of motions per hour increased over 30, the proposed algorithm had a correlation of 0.83 and a precision of 1.65 beats per minute compared to the conventional statical algorithm of a correlation of 0.56 and a precision of 4.04 beats per minute. Authors conclude that developed algorithm could reduce motion artifacts and improve heart rate and cardiac output monitoring accuracy and reliability, particularly in high-motion environments.

The title describes the core message of the paper.

The abstract incorporates key messages, in a concise manner.

The structure of the paper is accurate.

Conclusions are consistent with the evidence and arguments presented.

There are many Figures that make the article easier to read.

However, I have one major suggestion regarding this paper:

However, as is mentioned in limitation section, it is important to note that this algorithm is unsuitable for use in datasets that do not have good-quality signals with source consistency. I'm afraid that in everyday clinical practice there may be problem with it. It should be precise in potentially how big group of patients this problem may exist (based on literature data and according to data obtained in Author's hospital).

Author Response

We greatly appreciate the reviewers for their efforts and valuable comments to improve the manuscript. Based on the reviewers’ comments and suggestions, we made major revisions. In the revised manuscript, we used “Tracking Changes” function to mark up the changed portions. Our detailed responses to the reviewers’ comments and questions are as follows in the doc and pdf files:

Reviewer 2 Report

In my opinion, is quite an interesting paper. This study proposes a new algorithm to detect and correct motion artifacts (Mas) in the cardiac-related volume signal (CVS) to monitor conscious patients’ heart rate (HR) and cardiac output (CO).

There are some  comments in the reviewer's opinion that should be taken under consideration by the Author:

1.    Please give limitations of your study

2.    Please add the date of approval of e Institutional Review Board of Kyung Hee University Hospital  

3.    The Authors examined only 14 patients and made only 36 records- whether it would be good to add to the title- a preliminary study?

4.    Please add the inclusion and exclusion criteria for patients

5.    Please explain all abbreviations in descriptions of figures

6.    Please add the subsection Material and methods-statistical analysis

7.    Please underline the advantages and disadvantages of the method

Author Response

The previous works have evaluated the performance of developing algorithms on simulation data or measurements from healthy subjects, which are usually very different from ill patients. In addition, most of the attempted MAs reduction algorithms are based on frequency analysis. Thereby, these methods may be less effective for detecting types of MAs with distortions in the signal shape.

Reviewer 3 Report

Dear All, 

Review for the Paper title: “Motion Artifacts Reduction for Noninvasive Hemodynamic Monitoring of Conscious Patients using Electrical Impedance Tomography”

Authors should consider the following necessary changes:

1)  In the abstract authors mentioned that “The developed algorithm could reduce MAs and improve HR/CO monitoring accuracy and reliability, particularly in high-motion environments”.

               My question by how much it can be improved?

2)     Authors need to add statements about the plan for future work.

3)     At the end of the introduction, authors should add a paragraph about the structure of the paper.

4)     More literature review needs to be added latest publications (2020-2023) and I do recommend the following two papers:

-Investigating Gelatine Based Head Phantoms for Electroencephalography Compared to Electrical and Ex Vivo Porcine Skin Models:

https://ieeexplore.ieee.org/document/9475461

-A mini-review of graphene based materials for electrodes in electrocardiogram (ECG) sensing

 https://ieeexplore.ieee.org/document/9469827

- Aristovich K. Opinion: the Future of Electrical Impedance Tomography. J Electr Bioimpedance. 2022 Mar 31;13(1):1-3. doi: 10.2478/joeb-2022-0001.

- S. Kohli and A. J. Casson, ‘‘Removal of gross artifacts of transcranial alternating current stimulation in simultaneous EEG monitoring,’’ Sensors,vol. 19, no. 1, p. 190, 2019 

https://pubmed.ncbi.nlm.nih.gov/30621077/

5)     Authors need to add paragraph at the end of literature review to highlight the main contribution and how the work different from others.

6)     Figure 4 need to be clearer so authors should regenerate these figures and preferable to make them colorful with high quality.  

7)     At the end of discussion part authors should add table to make a comparison between their study and other researchers’ contributions.

All the best

Author Response

(The authors gave the same response as above.)

Reviewer 4 Report

The experimental study is interesting information in this paper. However, the main weakness of the paper lies in its lack of originality and novelty. The following suggestions may be considered to enhance the quality and clarity of the manuscript.

1-   Abstract is comprehensive and well written, but it needs improvements e.g. case study can be described a little more at the end of the paragraph.

2-   some state-of-the-art papers on medical imaging should be taken into account are:

https://www.sciencedirect.com/science/article/pii/S0020025522007332

https://www.sciencedirect.com/science/article/pii/S0957417422024940

3-   In experiments, the experimental analysis seems insufficient, which can not verify the motivation and contributions of this work.

4-   The dataset used can be summarized in a table format

5-   The paper only lists existing works in the research community without any analysis of existing work's limitations. Therefore, I suggest that the authors mention more summary and limitation analysis so that readers can easily appreciate the contributions made by this paper.

6-   Most of the references cited in the manuscript are old fashioned, the authors should ensure that all references cited in the manuscript are up-to-date and relevant to the research topic.

7-   In addition to these specific recommendations, the authors should also run the manuscript through a grammar checker like Grammarly to address any language or grammatical errors

the authors should also run the manuscript through a grammar checker like Grammarly to address any language or grammatical errors.

Author Response

(The authors gave the same response as above.)

Round 2

Reviewer 1 Report

Dear Authors and Editirs, 

The paper is improved now. I have no additional suggestions.

Reviewer 3 Report

Dear Authors, 

Thanks for considering all comments, the paper improved significantly. 

Reviewer 4 Report

The authors addressed all the comments thoroughly, thanks.